# Interdisciplinary perspectives on multimorbidity in Africa: Developing an expanded conceptual model

Justin Dixon[1,2]*, Ben Morton[3,4], Misheck J. Nkhata[5], Alan Silman[6], Ibrahim G. Simiyu[4], Stephen A. Spencer[3,4], Myrna Van Pinxteren[7], Christopher Bunn[8,9‡], Claire Calderwood[1,10‡], Clare I. R. Chandler[2‡], Edith Chikumbu[8‡], Amelia C. Crampin[8,11,12‡], John R. Hurst[13‡], Modou Jobe[14‡], Andre Pascal Kengne[15‡], Naomi S. Levitt[7‡], Mosa Moshabela[16‡], Mayowa Owolabi[17‡], Nasheeta Peer[15‡], Nozgechi Phiri[8‡], Sally J. Singh[18‡], Tsaone Tamuhla[19‡], Mandikudza Tembo[1,10‡], Nicki Tiffin[19‡], Eve Worrall[20‡], Nateiya M. Yongolo[4,21‡], Gift T. Banda[3,4¶], Fanuel Bickton[3,22¶], Abbi-Monique Mamani Bilungula[23¶], Edna Bosire[24,25¶], Marlen S. Chawani[3,26¶], Beatrice Chinoko[3¶], Mphatso Chisala[8¶], Jonathan Chiwanda[27¶], Sarah Drew[28¶], Lindsay Farrant[29¶], Rashida A. Ferrand[1,10¶], Mtisunge Gondwe[3,4¶], Celia L. Gregson[1,28¶], Richard Harding[30¶], Dan Kajungu[31¶], Stephen Kasenda[8¶], Winceslaus Katagira[32¶], Duncan Kwaitana[33¶], Emily Mendenhall[34¶], Adwoa Bemah Boamah Mensah[35¶], Modai Mnenula[36¶], Lovemore Mupaza[37¶], Maud Mwakasungula[38¶], Wisdom Nakanga[8,39¶], Chiratidzo Ndhlovu[40¶], Kennedy Nkhoma[30¶], Owen Nkoka[8,9¶], Edwina Addo Opare-Lokko[41¶], Jacob Phulusa[3¶], Alison Price[8,11¶], Jamie Rylance[3,4¶], Charity Salima[42¶], Sangwani Salimu[3,4¶], Joachim Sturmberg[43,44¶], Elizabeth Vale[45¶], Felix Limbani[3]*

**Data Availability Statement:** Source materials have been deidentified and deposited in Harvard Dataverse. These include the rapporteur notes from the concept-building workshop upon which

1 The Health Research Unit Zimbabwe, Biomedical Research and Training Institute, Harare, Zimbabwe, 2 Department of Global Health and Development, London School of Hygiene and Tropical Medicine, London, United Kingdom, 3 Malawi-Liverpool-Wellcome Programme, Blantyre, Malawi, 4 Department of Clinical Sciences, Liverpool School of Tropical Medicine, Liverpool, United Kingdom, 5 SHLS Nursing and Midwifery, Teesside University, Middlesborough, United Kingdom, 6 Nuffield Department of Orthopaedics, Rheumatology and Musculoskeletal Sciences, Oxford University, Oxford, United Kingdom, 7 Faculty of Health Sciences, Department of Medicine and Chronic Disease Initiative for Africa, University of Cape Town, Cape Town, South Africa, 8 Malawi Epidemiology and Intervention Research Unit, Lilongwe, Malawi, 9 College of Social Sciences, University of Glasgow, Glasgow, Scotland, United Kingdom, 10 Department of Clinical Research, London School of Hygiene & Tropical Medicine, London, United Kingdom, 11 Department of Population Health, London School of Hygiene & Tropical Medicine, London, United Kingdom, 12 School of Health and Wellbeing, University of Glasgow, Glasgow, United Kingdom, 13 UCL Respiratory, University College London, London, United Kingdom, 14 MRC Unit The Gambia at LSHTM, Banjul, The Gambia, 15 Non-communicable Diseases Research Unit, South African Medical Research Council, Cape Town, Durban, South Africa, 16 School of Nursing and Public Health, University of KwaZulu-Natal, Durban, South Africa, 17 Centre for Genomic and Precision Medicine, University of Ibadan, Ibadan, Nigeria, 18 Department of Respiratory Sciences, University of Leicester, Leicester, United Kingdom, 19 South African National Bioinformatics Institute, University of the Western Cape, Cape Town, South Africa, 20 Department of Vector Biology, Liverpool School of Tropical Medicine, Liverpool, United Kingdom, 21 Kilimanjaro Clinical Research Institute, Moshi, Tanzania, 22 Department of Rehabilitation Sciences, The Kamuzu University of Health Sciences, Blantyre, Malawi, 23 Department of Physical Medicine and Rehabilitation, University of Kinshasa, Kinshasa, Democratic Republic of Congo, 24 Brain and Mind Institute, Aga Khan University, Nairobi, Kenya, 25 SAMRC Developmental Pathways for Health Research Unit, University of the Witwatersrand, Johannesburg, South Africa, 26 Health Economics and Policy Unit, The Kamuzu University of Health Sciences, Blantyre, Malawi, 27 Department of Non-communicable Diseases, Ministry of Health, Lilongwe, Malawi, 28 Musculoskeletal Research Unit, Translational Health Sciences, Bristol Medical School, University of Bristol, Bristol, United Kingdom, 29 Faculty of Health Sciences, School of Public Health and Family Medicine, University of Cape Town, Cape Town, South Africa, 30 Florence Nightingale Faculty of Nursing Midwifery and Palliative Care, Cicely Saunders Institute, King's College London, London, United Kingdom, 31 Makerere University Centre for Health and Population Research, Makerere University, Kampala, Uganda, 32 Makerere Lung Institute, Makerere University, Kampala, Uganda, 33 Department of Family Medicine, The Kamuzu University of Health Sciences, Blantyre, Malawi, 34 Edmund A. Walsh School of Foreign Service, Georgetown University, Washington, DC, United States of America, 35 Department of Nursing, College of

this article is based and are available upon reasonable request. The DOI for the source materials in the repository are: https://doi.org/10.7910/DVN/YVO7SW.

**Funding:** This workshop on which this research was based was funded by the National Institute of Health and Care Research [Multilink Consortium, grant ref. NIHR201708, to FL, BM, JR, EW, IGS, NMY, GTB, SS) and the NIHR Leicester Biomedical Research Centre (BRC), to SJS], The Wellcome Trust [Multimorbidity and Knowledge Architectures: An Interdisciplinary Global Health Collaboration, grant ref. 222177, to JD] and the Department of Respiratory Sciences, University of Leicester, UK, supported by the British Academy and Global Challenges Research Fund [grant ref. GCRFNGR5\1242, to SJS]. The funders had no role in study design, data collection and analysis, decision to publish, or preparation of the manuscript.

**Competing interests:** The authors have declared that no competing interests exist.

Health Sciences, Kwame Nkrumah University of Science and Technology, Kumasi, Ghana, **36** College of Medicine, University of Malawi, Blantyre, Malawi, **37** Island Hospice and Healthcare, Harare, Zimbabwe, **38** Malawi NCD Alliance, Lilongwe, Malawi, **39** Deanery of Clinical Sciences, College of Medicine and Veterinary Medicine, University of Edinburgh, Edinburgh, United Kingdom, **40** Internal Medicine Unit, Faculty of Medicine and Health Sciences, University of Zimbabwe, Harare, Zimbabwe, **41** Greater Accra Regional Hospital, Faculty of Family Medicine, Ghana College of Physicians and Surgeons, Accra, Ghana, **42** Achikondi Women and Community Friendly Health Services, Lilongwe, Malawi, **43** School of Medicine and Public Health, Faculty of Health and Medicine, University of Newcastle, Newcastle, Australia, **44** International Society of Systems and Complexity Sciences for Health, Waitsfield, VT, United States of America, **45** University of the Witwatersrand, Johannesburg, South Africa

осε These joint first authors (listed alphabetically) contributed equally to this work.
‡ This article working group (listed alphabetically) contributed equally to this work
¶ This workshop collaborator group (listed alphabetically) contributed equally to this work
* justin.dixon@lshtm.ac.uk(JD); flimbani@mlw.mw(FL)

## Abstract

Multimorbidity is an emerging challenge for health systems globally. It is commonly defined as the co-occurrence of two or more chronic conditions in one person, but its meaning remains a lively area of academic debate, and the utility of the concept beyond high-income settings is uncertain. This article presents the findings from an interdisciplinary research initiative that drew together 60 academic and applied partners working in 10 African countries to answer the questions: how useful is the concept of multimorbidity within Africa? Can the concept be adapted to context to optimise its transformative potentials? During a three-day concept-building workshop, we investigated how the definition of multimorbidity was understood across diverse disciplinary and regional perspectives, evaluated the utility and limitations of existing concepts and definitions, and considered how to build a more context-sensitive, cross-cutting description of multimorbidity. This iterative process was guided by the principles of grounded theory and involved focus- and whole-group discussions during the workshop, thematic coding of workshop discussions, and further post-workshop development and refinement. Three thematic domains emerged from workshop discussions: the current focus of multimorbidity on constituent diseases; the potential for revised concepts to centre the priorities, needs, and social context of people living with multimorbidity (PLWMM); and the need for revised concepts to respond to varied conceptual priorities amongst stakeholders. These themes fed into the development of an expanded conceptual model that centres the catastrophic impacts multimorbidity can have for PLWMM, families and support structures, service providers, and health systems.

## Introduction

Multimorbidity–commonly defined as two or more long-term conditions in one person [1]–has become increasingly common as life expectancies rise globally, presenting a profound challenge to the organisation of current health systems around single diseases [2]. While the bodies of literature around multimorbidity were initially weighted towards high-income countries (HICs), multimorbidity has more recently been recognised as a global health challenge that may be especially detrimental in low- and middle-income countries (LMICs) [2–4].

Priority-setting initiatives for responding to multimorbidity in a global context [3] including in sub-Saharan Africa specifically [5,6] highlight the need to identify common disease 'clusters' and their shared determinants; develop integrated prevention and management approaches to jointly address multimorbidity clusters; and to more broadly to restructure health systems to become more holistic and person-centred.

With multimorbidity a growing research priority in Africa, the challenges it presents in this setting is becoming clearer. Like many intractable health system challenges, multimorbidity can be linked to centuries of colonialisation and racial discrimination that have systematically limited access to educational opportunities, employment, adequate housing, and basic health-care [7,8]. Chronic infectious diseases including HIV and tuberculosis (TB) that have thrived in such contexts increasingly intersect with non-communicable diseases (NCDs) including diabetes, cardiovascular diseases, and mental illness which have sharply risen due to the expansion of HIV treatment coverage, rising life expectancies, and lifestyles of poverty and urbanisation [7,9–11]. The resulting multimorbidity burden has been characterised in terms of the 'colliding epidemics' of chronic infectious and NCDs, generally affecting a younger, more economically-active demographic than in HICs, socially patterned along the lines of intersectional inequalities [4,12]. Most people living with multimorbidity (PLWMM) rely on public sector care, which generally remains donor-dependent, siloed, and heavily biased towards HIV and TB, with very limited funding available for NCDs [4,7]. There remains a stark disparity in quality of care for people with NCDs and, despite ongoing efforts, limited progress with integrating NCD care with programmes for chronic infectious diseases [13–15]. Fragmented, unevenly-resourced systems place a heavy health, social, and economic on burden on PLWMM, families, and wider social networks, undermining abilities to self-manage conditions and heightening the risk of complications and further morbidities [16,17].

Just as understandings of multimorbidity in Africa are advancing, it is also becoming clearer that multimorbidity means different things to different stakeholder groups. This reflects the diversity of disciplines, perspectives, methods, measurement instruments, and geographic vantage points from which they enter this emerging field. Indeed, despite its apparent simplicity and widespread use, the definition of multimorbidity as two or more long-term conditions endorsed by the WHO [1] and Academy of Medical Sciences [3] remains contested. Debates remain as to how many and which conditions 'count' as multimorbidity; [18,19] whether it should be expanded to include causes and consequences as well as presence/ absence; [6] and indeed, whether the very concept reflects a culture of super-specialised biomedicine in the global North that may not translate well within low-resource settings [8,20,21]. Consequently, it remains challenging to compare multimorbidity across datasets, to communicate across disciplines, and to gauge the utility and possible harms of applying multimorbidity as a concept within African settings. As commentators have noted, dominant biomedical discourses have been centred on the search for a universal definition, assumed to be a prerequisite for action, but this has yielded poor returns [20,22]. Others have argued that seeking to universalize multimorbidity misses the point that its power may lie precisely in its resistance to being pinned down to a number or specification of conditions, forcing us instead to consider the whole person in context [20]. This is a compelling proposition that compels the development of concepts and models guided by the specificities of context rather than by the (perhaps futile) imperative to universalize multimorbidity. Yet without a common lexicon that enables communication across different perspectives, we may miss the opportunity to build on the current momentum around multimorbidity to maximise benefits to patients and their carers across geographies, incomes and societal structures.

Responding to this need, this article presents the findings and outcomes from an interdisciplinary research initiative to interrogate the conceptual underpinnings of multimorbidity

research and care in Africa. It sought to address the questions: how useful is the concept of multimorbidity within the African context? Can the concept be adapted for it to be more context-sensitive, cross-cutting, and more transformative? Initiated during a three-day concept-building workshop in Blantyre, Malawi, participants collaboratively explored the meaning of multimorbidity across different disciplines and perspectives, critically appraised their potential, limitations, and utility; and iteratively pieced together a cross-disciplinary conceptual model designed to be sensitive to the particularities and heterogeneity of the African context. By folding a wide range of disciplinary perspectives, concerns, and interests into a common framework, the resultant multimorbidity model can, we contend, underpin and orient a cross-disciplinary, Southern-led response to multimorbidity in Africa.

## Methods

### Research design

We used an inductive, co-productive research design influenced by principles of participatory research and constructivist grounded theory [23]. The research process involved a three-day concept-building workshop in Blantyre, Malawi (June 22–24, 2022), [21] thematic 'open coding' of workshop discussions, and further iterative development of a conceptual model following the workshop. Our approach follows the growing interest in "collective experimentation" [24] to address issues of transdisciplinary concern in public and global health, [24–26] of which multimorbidity is arguably a paradigmatic example.

### Participants and sampling

The workshop organising committee comprised an interdisciplinary group of public health researchers, clinicians, and social scientists (EB, CIRC, JD, RAF, FL, EM, BM, JR). Potential participants were identified through purposive and snowballing methods between June 2021-March 2022, which are described in greater detail elsewhere [21]. The collaborator group drew together 60 researchers, clinicians, health planners, and policymakers (HIC-based n = 19, LMIC-based (n = 41), together representing a wide range of disciplinary perspectives, including from (sub-)fields of epidemiology, public health, clinical medicine, the social sciences and community advocacy (Fig 1). Given the current concentration of multimorbidity research, and the location of the workshop in Malawi, the regional expertise among the collaborator group stemmed primarily from countries within Southern Africa, with the greatest concentration of experience from Malawi (n = 33), South Africa (n = 13), and Zimbabwe (n = 9). Participants, several of whom were working across multiple countries, also brought experience from Central, Eastern, and Western Africa, together representing experience from 10 African countries (Fig 1). For practical and ethical reasons, we did not directly include PLWMM in the workshop; however, we asked collaborators, including social scientists and community advocates present, to represent learnings from their interactions to give an understanding of different patient perspectives. Supporting information includes a reflexivity statement detailing the measures taken to promote equitable partnership within this collaboration (S1 Table), [27] an overview of participant demographics (S2 Table), and a detailed breakdown of participants' institutional location and disciplinary and regional expertise (S3 Table).

### Workshop design

A detailed account of the workshop design is published elsewhere [21]. The workshop was designed to optimise opportunities for cross-disciplinary discussion. Sessions were organised around four provisional thematic 'domains': (1) concepts and framings of multimorbidity (led

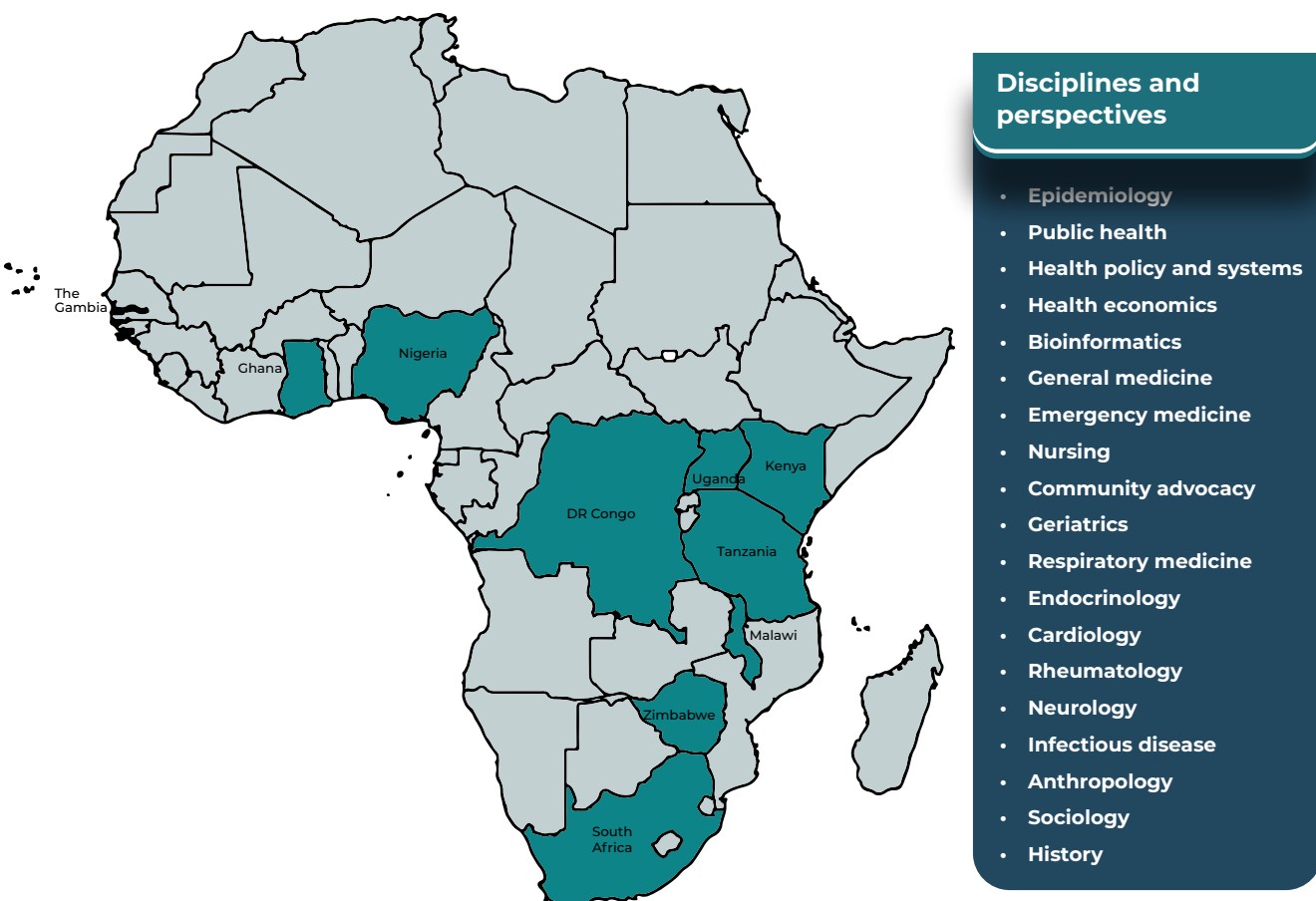

**Fig 1. Regional\* and disciplinary expertise represented by workshop participants.** \*Base map produced using R[28] with shape files obtained from the World Bank (https://datacatalog.worldbank.org/search/dataset/0038272/World-Bank-Official-Boundaries) and Humanitarian Data Exchange (for Western Sahara only) (https://data.humdata.org/dataset/cod-ab-esh).

by EC and CB); (2) population-level health data (led by NT and TT); (3) risk, prevention, and sites of intervention (led by NP and APK); (4) health systems and care models (led by EB). Each session began with an 'ignition' talk by session leads which outlined current knowledge, gaps, and key questions within each domain. These questions were then addressed by participants through a plenary discussion, facilitated by session leads and workshop organisers, and smaller focus-group discussions. The latter were facilitated by a team of PhD student rapporteurs (GTB, SS, IGS, SAS, NMY), who were mentored by members of the organising committee. In a final session, the group reviewed and summarised the workshop sessions in plenary, collectively identifying core and cross-cutting themes, before breaking into working groups which advanced agreed themes for further development. This article was led by working group focused on developing a common concept of multimorbidity.

## Analytical framework

This research was guided by the principles of constructivist grounded theory, which holds that there is no 'right or wrong' and that emergent concepts and grounded theories are interpretive descriptions rather than an 'objective' account of reality [23]. Accordingly, the assumption underlying the workshop was that there is no privileged disciplinary vantage point from which

to conceptualise or frame multimorbidity. Rather, different concepts and understandings fore-ground different aspects of the challenge multimorbidity presents for current systems, tied to particular knowledge bases, while potentially backgrounding others. This position, which was emphasised throughout the workshop and sustained through the post-workshop concept development, encouraged a diversity of concepts and understandings to be put into conversa-tion, some of which diverged from and exposed the limits of the prevailing biomedical model. At the same time, in embracing difference, we were also able to find synergies and shared com-mitments. The embrace of both diversity and commonality formed the basis of developing a more cross-cutting, holistic, and ultimately more useful understanding of multimorbidity as it relates to African contexts.

## Data collection

Source material collected during the workshop included, most centrally, detailed notes of pro-ceedings, including both focus groups and plenary discussions, taken by the rapporteur team. Rapporteurs were instructed to capture summaries of what speakers had said and, where points were made that the rapporteurs felt were especially significant, direct quotes were also taken. Where necessary, the rapporteurs went back to individuals for clarification on points made, which were absorbed into their notes. Also absorbed into the final account were the flip chart pages composed during focus groups and the Microsoft Word documents produced using a shared screen during plenary discussion. Finally, all source materials were reconciled to produce a unified account of both focus-group and plenary discussions. These were made available to all in unanonymised form through a secure online storage platform, before being anonymised in the final deposited dataset.

## Data analysis

As this was an iterative, co-productive process, [23] analysis began during the workshop itself, with all participants in engaging in critical reflection during sessions and collaboratively draw-ing out major cross-cutting issues in the final session [21]. After the rapporteurs had produced the final account of proceedings, the joint first authors (JD, BM, MJN, AS, IGS, SAS, MVP) subsequently conducted iterative inductive thematic analysis on the data using a team-based, 'open coding' approach that is designed to optimise inter-coder consensus in the context of collaborative research [29] To facilitate this process, data were transferred onto a shared drive, which enabled the coding team to code collaboratively in real time. After an initial pass through the data, the team met to consolidate initial codes into a provisional framework, which was subsequently applied to the whole dataset by JD and MVP. A second meeting was used to further refine the framework, with an emphasis on agreeing higher-level themes that grouped and explained lower-level ones. The final framework, of which the (sub-)themes related to the conceptualisation and framing of multimorbidity are presented in the results sec-tion, was then taken to the larger working group and used to develop of an expanded model of multimorbidity. Developing and refining this model was itself an iterative process, involving the working group assembling a tentative visual model based on the coding framework and relevant literature, which was circulated by email to the wider collaborator group along with the draft manuscript. Feedback was used to refine the model before being sent back for a final round of comment and refinement.

## Research ethics

Due to the collaborative research design employed, in which all investigators were participants and vice versa, and all are named co-authors, formal ethical review was not required for this

research. This follows recent examples adopting similar co-productive research models to develop cross-disciplinary frameworks and agendas [25,30]. All participants provided either verbal consent or written email consent to taking part in the concept-building workshop, which was captured using an Excel spreadsheet. At the end of the workshop, participants jointly agreed upon research outputs and working groups to take forward prominent themes from the discussions, and all consented to being named as co-authors on this particular output. All participants in the initiative have reviewed the contents of the manuscript and have approved its final version. The concept-building workshop was subject to the safeguarding mechanisms of the host organisation (Malawi-Liverpool-Wellcome Programme), which included anonymous reporting procedures.

## Results

Three overarching themes representing the group's shared commitments for the conceptualisation of multimorbidity in Africa emerged: (1) the current focus of multimorbidity on constituent diseases; (2) the potential for revised concepts to centre the priorities, needs, and social context of PLWMM; (3) and the need for revised concepts to respond to varied conceptual priorities amongst stakeholders." These themes and their constituent sub-themes are summarised in Table 1.

### Theme 1: Multimorbidity is currently centred on constituent diseases

Across the countries represented at the workshop, a key challenge expressed by participants was that populations are living longer and facing increasingly complex disease burdens, including the colliding epidemics of chronic infectious diseases and NCDs. Yet, health systems

**Table 1. Key themes and sub-themes for conceptualising multimorbidity.**

| Theme | Sub-Themes |
|---|---|
| Multimorbidity is currently centred on constituent diseases | • In medicine and global health, patients are defined by their diseases<br>• The minimalist definition of multimorbidity as two or more long-term conditions promotes a disease-centric view<br>• This definition also promotes narrow consideration of social factors associated with multimorbidity centred on individual behaviour and lifestyle factors<br>• Interventions working with this definition may continue to reinforce or exacerbate the status quo |
| Potential for revised concepts to centre the priorities, needs, and social context of PLWMM | • Benefit of focusing less on the disease categories within multimorbidity and more on its common consequences for PLWMM and associated needs<br>• The burden of multimorbidity on families, informal carers, and social networks is crucial to consider in many African contexts<br>• Need for a broader appreciation of the social, structural, and environmental context of multimorbidity<br>• Fragmented, disease-driven systems compound the burden of multimorbidity |
| Need for revised concepts to respond to varied conceptual priorities amongst stakeholders | • A standard, one-size-fits-all definition not a panacea for multimorbidity<br>• Multimorbidity may mean different things, carry different priorities, and be more or less useful in different contexts and at different levels of scale<br>• A broad, flexible understanding that recognises multiple perspectives is needed to be cross-cutting and useful |

remain built around siloes of expertise based on single diseases. This is perpetuated by 'vertical' funding models adopted by Northern donors, which work against care for multimorbidity:

"Most of the chronic diseases in sub-Saharan Africa countries such as HIV and TB are managed through vertical programs, which inhibits care for multimorbidity" (Session 1, plenary, rapporteur notes)

Vertical approaches to single diseases have come to shape almost all aspects health system functioning: from health policy and planning, to research and surveillance, to training and care delivery, to the monitoring and evaluation of health systems performance. Disease-centred thinking is, as one family physician argued, so entrenched within current systems that it is extremely challenging to step beyond this frame of reference:

"The challenge is that the system we work in does not accommodate a transformation in thinking; it keeps bringing us back to a focus on diseases" (Session 1, plenary, rapporteur notes)

Throughout the workshop, participants grappled with the tension that, on the one hand, multimorbidity foregrounds disease concentrations and interactions rather than diseases in isolation. This makes it a potentially powerful concept for re-aligning priorities and systems with the increasingly complex disease burdens affecting many African countries. On the other hand, the concept of multimorbidity, participants noted, is generally that of a compound disease category, most commonly defined as the presence or absence of two or more long-term conditions. As a result, it is still a disease-centred concept. Participants therefore questioned whether this definition was suited for moving us from a scenario in which people are defined by their diseases, to what was often referred to as a more 'person-centred' approach:

"The definition of two or more conditions may have limitations, and may continue to perpetuate a disease specific approach" (Session 5, plenary, Word document on shared screen)

"We are still using a disease-focused lens to define multimorbidity–tensions versus a person-centred approach" (Session 5, group 5, flip chart excerpt)

Concerns raised about a disease-centred lens included that this may promote an overly simplistic, additive view of multimorbidity as the sum of interacting (but ultimately discrete) disease conditions. Relatedly, reducing multimorbidity to its constituent diseases was argued to narrow consideration social context to 'modifiable lifestyle factors' associated with particular conditions, a particularly prominent discourse in the context of NCDs (i.e., smoking, poor diet, sedentism, and substance abuse). As several participants noted, targeting individual behaviour and lifestyles does not amount to a 'person-centred' approach to multimorbidity and may lead to patient shaming and stigmatisation:

"Labelling some of the non-communicable conditions related to multimorbidity, such as hypertension and diabetes, as 'lifestyle diseases' creates stigma and shaming" (Session 3, group 2, rapporteur notes)

In turn, this behavioural and lifestyle focus deemphasises structural factors that constrain people's lifestyle 'choices' in ways that make them vulnerable to multimorbidity.

### Theme 2: Potential for revised concepts to centre the priorities, needs, and social context of PLWMM

Continuous with theme 1, a particular concern expressed with the disease-based definition of multimorbidity is that it also fails to capture what *matters* to PLWMM. Several points about patients' needs and priorities were highlighted. First, a concern voiced particularly by clinicians was that it often matters far less to people what the diagnoses and their causes are than their consequences and impacts, which become especially complex when it comes to secondary medical complications (e.g.. diabetic nephropathy). Symptom and treatment burden, functionality, and quality of life were highlighted as important to fold into our understanding of multimorbidity:

> "Multimorbidity is likely to be useful if it includes burden and function, for instance pain, disability, and sleep" (Session 1, group 1, rapporteur notes)

While there may be considerable heterogeneity in the lived experience of multimorbidity across different condition combinations, it was also observed that the symptoms, treatment burden, and needs of PLWMM are often not disease-specific and may share similar profiles regardless of condition or aetiology. Tellingly, when it was put to a vote whether the concept of multimorbidity should draw a distinction between 'communicable' and 'non-communicable' diseases, a significant majority said it should not. Participants were not suggesting that we cease thinking about causes and determinants of multimorbidity, which remain important especially within epidemiology, public health, and clinical medicine, but rather that we ground our concept of multimorbidity in the experiences and priorities of PLWMM.

Second, participants emphasised that when considering the impacts of multimorbidity in African contexts, it is vital to expand the focus beyond the affected person to the pivotal role of families, informal caregivers and larger support networks in navigating the health, social, and economic burdens of multimorbidity:

> "Family members/community are very important for patients' improvement because they have an influence on the treatment given, in the same way that nutrition post-delivery is mostly influenced by relatives" (Session 1, group 3, rapporteur notes)

Accordingly, participants including community advocates argued that while person-centred care is important, we need to be talking about *family*-centred care in this setting:

> "We need to think about family centred care–patient-centred care is important, but there is also a family who has to be involved in the process" (Session 5, plenary discussion, Word document on shared screen)"

A related consideration when designing multimorbidity care models for African health systems is not to ignore current social realities such as medical pluralism and the existing role of traditional healers. In the absence of such recognition, care models risk recreating Northern global health policies which fail to account for local experiences and African-centred knowledge systems and values.

Third, widening the lens further still, participants argued that social, structural and environmental factors are crucial for understanding the patterning, impacts, and consequences of multimorbidity. The concepts of syndemics and the social and commercial determinants of health were recognised as useful and well-established frameworks for capturing 'upstream' factors that socially pattern multimorbidity along lines of poverty and inequality:

"Social determinants of health are important–syndemics, the commercial determinants of health, and others. For example, for TB patients, food insecurity, biological predisposition, and access to care have been used to identify patients/potential patients with multimorbidity" (Session 3, group 2, rapporteur notes)

Social scientists reported from qualitative research that daily socio-economic struggles preceded and exacerbated bodily ailments and symptoms, and that within such fragile arrangements, interruptions to or changes in circumstance could have huge impacts on overall ability to cope, as was evidenced during the COVID-19 pandemic. It was further observed that specialised care organised around single diseases compounds the burden placed on PLWMM: it shapes which diseases are diagnosed and prioritised, the time and resources that are needed to (self-)manage different conditions, and the added care burden on families and carers:

"Most care in urban areas is specialized. This leads to fragmentation, adding a burden to the family and caretakers" (Session 1, group 3, rapporteur notes)

"Individual disease treatment compounds the burden. Patients start prioritising certain conditions and some may go untreated" (Session 1, group 2, rapporteur notes)

Challenges commonly observed during qualitative research included reaching secondary and tertiary health facilities to receive care, especially for NCDs; struggles to mobilise funds to spend on multiple medicines and appointments; and losing income when spending time at the clinic or balancing medical appointments with family responsibilities.

## Theme 3: Need for revised concepts to respond to varied conceptual priorities amongst stakeholders

The third theme was that, if context is taken seriously, a one-size-fits-all definition of multimorbidity trying to pin down multimorbidity to a specific number or combination of conditions is neither possible nor desirable:

"There is no one-size-fits-all concept or framing of multimorbidity–context matters" (Session 5, group 1, flip chart)

The diversity of academic and applied perspectives present at the workshop underscored how counterproductive, even harmful, a narrow definition favouring one discipline or perspective can be. The concept, it was agreed, needs to remain broad, flexible, and able to foreground different priorities depending on the question being asked, the problem or perspective driving the question, the geographical, epidemiological, and health system context, and indeed the level of scale. Depending on the context, multiple definitions may be needed:

"Some flexibility / ambiguity in definition–or multiple definitions–may be needed" (Session 5, plenary discussion, Word document on shared screen)

Participants considered the utility of multimorbidity to different disciplinary and stakeholder groups, and how the priorities might shift in different contexts. As highlighted in Theme 2, the primary concern at the clinical and community level is functionality, quality of life, and the burden and complexity shouldered by PLWMM, their family and social networks:

"For academics, policymakers, and public health, the label or definition of multimorbidity is likely to be useful, whereas, for patients, wellbeing and function are more important" (Session 1, group 1, rapporteur notes)

Public health researchers, while similarly recognising the need for greater emphasis on function and quality of life, were often more focused on the societal impact of multimorbidity, for which well-defined (though not necessarily disease-centric) measures are needed to facilitate comparison:

"Well-defined concepts are useful in academia. The label [of multimorbidity] is useful for prevalence/mapping clusters, being able to study interactions between drugs and conditions" (Session 1, group 1, rapporteur notes)

For policymakers and health planners, also deploying multimorbidity at the population level, the concept was viewed as useful for reconfiguring funding streams (e.g., from 'vertical' to 'horizontal' models), for developing new care delivery models (e.g., from disease- to person- and family-centred care), for training and deploying the health workforce, and managing risks among the population. Social scientists and historians pushed for an expansive concept, one connecting the intricacies of lived experience to the 'upstream' structural and systemic factors that socially pattern multimorbidity and exacerbate the burden:

"You can't do multimorbidity and focus only on medical conditions, there are a lot of things are going on–social, financial, medical. These factors work together and need to be addressed together, including clinical interventions, upstream solutions, downstream solutions, and community interventions." (Session 4, ignition talk, rapporteur notes)

Finally, while PLWMM were not represented directly, the group noted that multimorbidity was unlikely to translate well into lay models:

"It is important to distinguish the medical framework and the patient model. Biomedical and societal framings of symptoms translate poorly into lay terminology–there is not necessarily a term for multimorbidity" (Session 5, plenary discussion, Word document on shared screen)

Because of these different concerns, we may need to foreground (and background) different aspects of multimorbidity, and recognise scenarios when it is not useful, to maximise its potential and minimise its harms:

"Part of the 'art' of multimorbidity may be centring diseases, people, and systems at different times and in different places and situations" (Session 5, group 1, rapporteur notes)

While recognising the impossibility of a one-size-fits-all definition given that multimorbidity is, by nature, heterogenous and context-specific, the collaborator group was nonetheless optimistic about folding the multiple perspectives represented at the workshop into a broadly shared frame of reference to facilitate cross-disciplinary working. Following theme 2, it was proposed a focus on common consequences of multimorbidity and associated needs for PLWMM could be one pathway towards conceptual alignment. In the final session of the workshop, a working definition was proposed as:

"A clustering of *needs* [added emphasis] and conditions that need to be addressed holistically, rather than in isolation" (Session 5, plenary discussion, Word document on shared screen).

Whilst only a starting point, it did draw together major points of agreement during the workshop: the foregrounding of clusters of needs rather than just medical conditions; the importance of a holistic purview; and the critique of compartmentalised approaches. Refining this concept was highlighted as a key aim moving forward.

## Discussion

Multimorbidity presents a major health system challenge for African countries. We identified three core themes that are pertinent to the conceptualisation of multimorbidity in Africa: the disease-centricity of current concepts; the potential for revised concepts to foreground what matters to PLWMM; and the need to accommodate varied conceptual priorities amongst different stakeholders. Building on this framework, in the following, we place these core themes into conversation with current global health scholarship on multimorbidity, thereby developing an expanded model of multimorbidity that can underpin a coordinated but context-specific response to multimorbidity in Africa.

Our analysis supports the conclusions of others who have argued that the current definition of multimorbidity as two-or-more-long-term conditions promotes a static, additive rendering of disease that treats illness as the sum of its parts and thus continues to define people by their diseases [20,31,32]. One consequence of this disease-centred framing, as Blarikom et al. [20] have argued, is that many researchers working within the field have become preoccupied and indeed, largely paralysed by the search for standard biomedical definition, which can ultimately undermine the concept's anticipated potential of moving us towards more person-centred research and care [20,31,32]. Recent calls to "decolonise" [33] multimorbidity highlight the need to recognise the 'colliding epidemics' of chronic infectious and NCDs that characterise multimorbidity in many low-resource settings, in contrast with the predominantly NCD-related multimorbidity in HICs [4]. While this may have important implications for integrating care across historically separated disease domains, the workshop proceedings suggest that decolonising multimorbidity means more than adjusting its constituent diseases. Rather, it means a more decisive shift away from models of research and care that define and categorise people by their diseases.

Such a shift, our analysis shows, requires foregrounding the priorities, needs, and social context of PLWMM. Calls for more holistic approaches resound in literature and align with theoretical perspectives from multiple disciplines. This includes eco-social theory from epidemiology; [34] the syndemic framework, [7,35] the theory of recursive cascades [36] and burden of treatment theory [17] from the social sciences, and novel applications of complexity theory to multimorbidity within the primary care sciences [31]. Despite this, attempts to integrate care have not proven to be especially 'person-centred' in practice. Current models, such as South Africa's Integrated Chronic Disease Management (ICDM) programme, have often been built based on vertical disease programmes and struggled to overcome legacies of system fragmentation, uneven resourcing of conditions, and individualized responsibility for self-management [14]. A growing number of qualitative studies reporting PLWMM's perspectives from Malawi, [17] Ghana [37], Ethiopia, [38] and South Africa [39] show that PLWMM are unable to take responsibility in self-management if they experience a lack of health services, information, and basic necessities. Such lack in turn feeds into a cycle of precariousness that negatively impacts people's ability to cope, often sending them down a slippery slope–or in complexity theory terms, over 'tipping points' [31]–towards further disability and decline.

Also appealed to but rarely prioritised in practice is the active involvement of caregivers and support networks in the provision of care for PLWMM [14,40]. Here, African social network theories such as Ubuntu–promoting mutual caring through compassion, reciprocity, dignity, and humanity–could be a useful concept, as they potentially cultivate resilience by creating a shared identity between PLWMM and carers, allowing them to flourish even when living in precarity [40]. The delivery of such care could involve the decentralisation of chronic care through health and social care workers trained in holistic, community-based care models–as elaborated, for instance, within the syndemic care framework [7,41].

Our findings suggest that a focus on common consequences of multimorbidity holds promise for bringing together different conceptual priorities of stakeholders. If, indeed, many of the needs and treatment burdens experienced by PLWMM are often not disease specific, such a disease-agnostic focus could bring together disciplines and specialities in a way that has eluded the abstract and seemingly irresolvable attempts to pin down multimorbidity and its causal pathways in pathophysiological terms. The broader point is that the distinction between causes and consequences itself begins to break down once we recentre the circumstances and trajectories of PLWMM, from whose perspective 'consequences' of a living with multimorbidity today may be 'causes' of further problems tomorrow [36]. By centring PLWMM and their ability to lead healthy, fulfilling and independent lives, we may better align medical and lay models of multimorbidity, such that it may not only be more cross-cutting and useful but also, perhaps, carry fewer negative connotations as a label. Fig 2 presents an expanded model of multimorbidity that folds in the findings of our analyses.

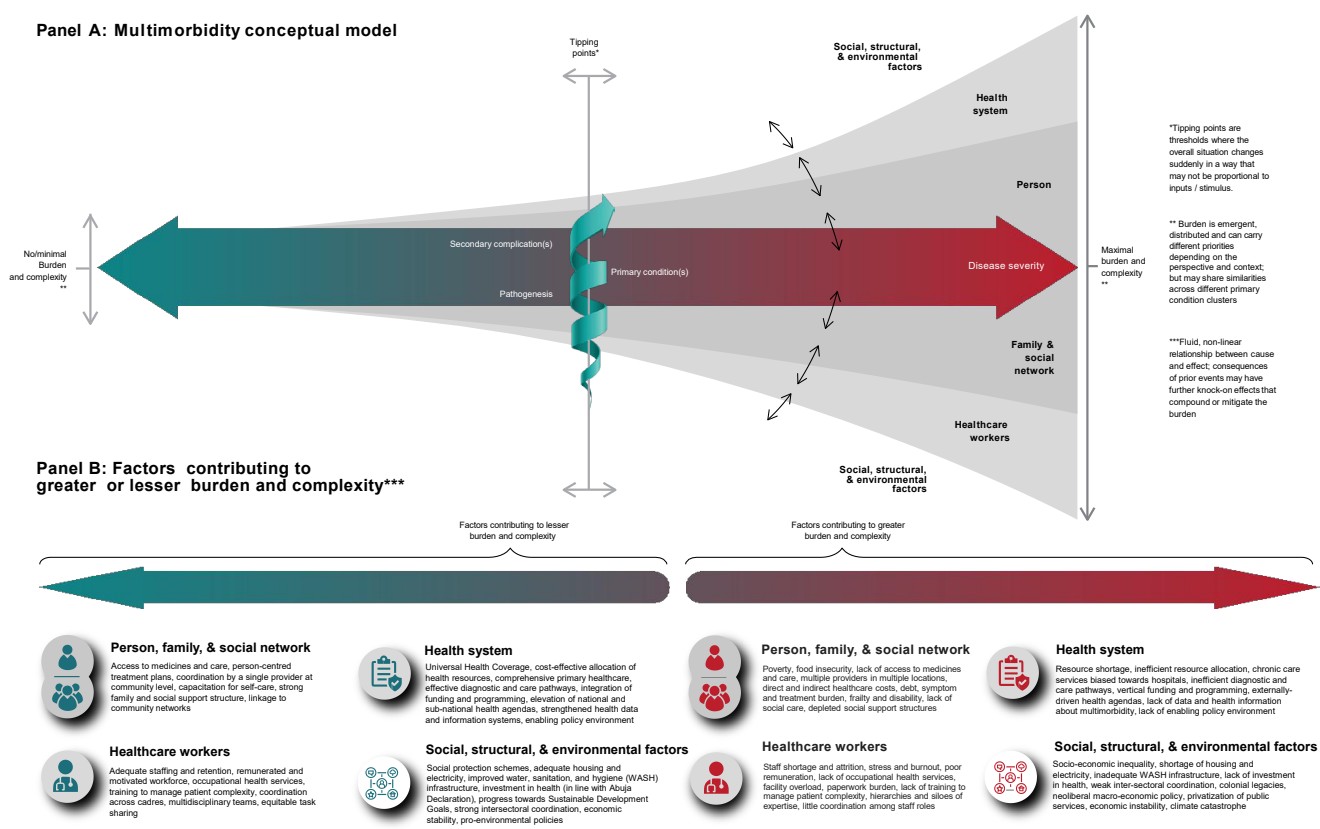

**Fig 2. An expanded conceptual model of multimorbidity.**

This model draws on the multiple theoretical influences of the participant group, including the syndemics framework, [7,35] burden of treatment theory, [17] and complexity theory [31]. It retains within it the basic idea of multimorbidity as involving more than one primary condition, in recognition of the utility of diagnostic categories within public health, epidemiology, and clinical medicine. But it remains agnostic about which kinds and combinations of condition 'count' as multimorbidity, instead expanding and bringing into the foreground the consequences of multimorbidity and its distributed burden on PLWMM, families, social networks, health providers, and the health system. The model highlights factors identified by the collaborator group that currently overwhelmingly pull towards the right of the model, that is, that set PLWMM on a slippery (but not inevitable or linear) slope towards secondary complications and potentially catastrophic burdens that are felt across the system. It also casts our attention to the factors that might pull back people towards the left of the model. This includes seemingly small changes to a person's circumstances that may make multimorbidity more manageable for the person affected and their wider networks, even in cases of advanced or complex disease. Given that many secondary complications cannot be reversed, the model suggests the emphasis should be on primary and secondary prevention. More than pharmacological or lifestyle interventions, this demands interventions that take into account the interdependency between PLWMM and family, social network, healthcare providers, the health system, and wider social, structural, and environmental context.

The dynamic and multidimensional nature of multimorbidity in this model enables different questions and disciplinary perspectives to be brought to bear on multimorbidity, including more refined definitions suited for specific questions. First, a focus on the complex bio-social interactions and impacts of multimorbidity moves us away from the disease-centred, cross-sectional designs from which most knowledge about multimorbidity to date originates (and that have tended to favour pharmacological and behavioural interventions) to disease-agnostic, richly contextual cohort research designs focused on burden, outcomes, and quality of life across the life course. Second, recognition of the multilevel factors at play implies that multimorbidity cannot be fully appreciated through any one disciplinary lens alone. Inter- and trans-disciplinary approaches, as well as holistic stakeholder engagement are needed to understand how biological, socio-cultural, political, economic, and environmental factors intertwine to co-produce multimorbidity; as well as to design holistic, person-centred, and systems-directed interventions. Third, the model implies a more context-specific approach to decide which disciplines and fields of expertise are relevant within the multimorbidity arena. Multimorbidity may result in similar needs and treatment burdens across different condition clusters, but its manifestation may also vary considerably across African settings given the tremendous heterogeneity of social, structural, environmental, and health system contexts. Thus, the knowledge bases and skillsets required to understand and address multimorbidity need to be similarly adaptive [8]. Whilst challenging, this is precisely the kind of context-sensitive approach that multimorbidity compels, offering a more promising pathway towards holistic person-centred care than universal, disease-centred interventions.

This model and the approach underlying it have several strengths. This was the first workshop to systematically bring together a multidisciplinary group of academic and applied actors to critically consider the meaning and utility of multimorbidity specifically within African settings. This was a bold exercise, not only in multidisciplinary experimentation, [24] but in elevating perspectives from a range of African contexts to reframe the multimorbidity conversation. This work provides a conceptual infrastructure to undergird the North-South and, more importantly, South-South partnerships that have been explicitly recognised as necessary for responding to multimorbidity in the region [5]. This research also had several limitations. First, while the collaborator group advocated for a person-centred perspective on

multimorbidity, we were unable to directly include the voices of PLWMM. Further research is needed to gauge the extent to which this model resonates with PLWMM, whose perspectives remain under-represented within the multimorbidity conversation in Africa. The collaborator group was also biased towards Southern Africa, with the largest representation of expertise from Malawi, Zimbabwe, and South Africa, and thus our work disproportionately reflects views from these countries. Finally, our thematic analysis may be biased towards the views of the core working group. Whilst we endeavoured to represent the wider group's perspectives objectively through a rigorous thematic analysis of workshop proceedings, the fact that we were both participants (in the workshop) and observers (analysing proceedings) means that the model may be biased towards our own values and viewpoints.

## Conclusion

In this article, we have analysed focus-and whole-group discussions from a workshop on multimorbidity in African contexts, placed emergent themes into conversation with current thinking on multimorbidity, and developed an expanded model based on the groups' common commitments. While recognising the crucial research that has preceded our work, we believe in the necessity of providing nuance to the available framings of multimorbidity, stressing the importance of understanding the lived experiences of PLWMM and their networks, and adding in the socio-economic complexities that impact PLWMM, providers, and the development of systems. Further conceptual and empirical work is needed to draw out the implications of this conceptual model for the heterogenous and multifaceted health systems in Africa. Also open to empirical scrutiny is whether it will prove useful for orienting and mapping different strands of multimorbidity work across disciplines, projects, and interventions, and for contributing to an overall more joined-up response moving forward.

## Supporting information

**S1 Checklist. COREQ checklist: Includes a completed COREQ checklist which is a recognised standard for reporting on qualitative studies.**
(DOC)

**S1 Table. Reflexivity statement.** Details measures taken during the research to promote equitable partnership within this collaboration.
(DOCX)

**S2 Table. Workshop participant summary table.** Provides an overview of gender balance, geographical representation, and career stage within the collaboration.
(DOCX)

**S3 Table. Full participant breakdown.** Describes each participant's institution(s) and disciplinary and regional expertise.
(DOCX)

## Acknowledgments

We are greatly indebted to members of the Africa Multimorbidity Alliance who took part in the concept-building workshop on which this article is based. We are grateful to the administrators and technicians at the Malawi-Liverpool-Wellcome Programme for their efforts in preparing and running the workshop, with particular thanks to Kate Mangulama. We finally wish to express gratitude to the many funders and institutions who supported the travel and attendance of participants.

## Author Contributions

**Conceptualization:** Justin Dixon, Ben Morton, Misheck J. Nkhata, Alan Silman, Ibrahim G. Simiyu, Stephen A. Spencer, Myrna Van Pinxteren, Christopher Bunn, Clare I. R. Chandler, Edith Chikumbu, Andre Pascal Kengne, Mosa Moshabela, Nasheeta Peer, Tsaone Tamuhla, Nicki Tiffin, Nateiya M. Yongolo, Gift T. Banda, Edna Bosire, Jonathan Chiwanda, Rashida A. Ferrand, Emily Mendenhall, Jamie Rylance, Sangwani Salimu, Felix Limbani.

**Data curation:** Ibrahim G. Simiyu, Stephen A. Spencer.

**Formal analysis:** Justin Dixon, Ben Morton, Misheck J. Nkhata, Alan Silman, Ibrahim G. Simiyu, Stephen A. Spencer, Myrna Van Pinxteren.

**Funding acquisition:** Justin Dixon, Ben Morton, Sally J. Singh, Felix Limbani.

**Investigation:** Justin Dixon, Ben Morton, Misheck J. Nkhata, Alan Silman, Ibrahim G. Simiyu, Stephen A. Spencer, Myrna Van Pinxteren, Christopher Bunn, Claire Calderwood, Clare I. R. Chandler, Edith Chikumbu, Amelia C. Crampin, John R. Hurst, Andre Pascal Kengne, Naomi S. Levitt, Mosa Moshabela, Mayowa Owolabi, Nasheeta Peer, Nozgechi Phiri, Sally J. Singh, Tsaone Tamuhla, Mandikudza Tembo, Nicki Tiffin, Nateiya M. Yongolo, Gift T. Banda, Fanuel Bickton, Abbi-Monique Mamani Bilungula, Edna Bosire, Marlen S. Chawani, Beatrice Chinoko, Mphatso Chisala, Jonathan Chiwanda, Sarah Drew, Lindsay Farrant, Rashida A. Ferrand, Mtisunge Gondwe, Celia L. Gregson, Richard Harding, Dan Kajungu, Stephen Kasenda, Winceslaus Katagira, Duncan Kwaitana, Emily Mendenhall, Adwoa Bemah Boamah Mensah, Modai Mnenula, Lovemore Mupaza, Maud Mwakasungula, Wisdom Nakanga, Chiratidzo Ndhlovu, Kennedy Nkhoma, Owen Nkoka, Edwina Addo Opare-Lokko, Jacob Phulusa, Alison Price, Jamie Rylance, Charity Salima, Sangwani Salimu, Joachim Sturmberg, Elizabeth Vale, Felix Limbani.

**Methodology:** Justin Dixon, Ben Morton, Misheck J. Nkhata, Alan Silman, Ibrahim G. Simiyu, Stephen A. Spencer, Myrna Van Pinxteren, Christopher Bunn, Clare I. R. Chandler, Edith Chikumbu, Andre Pascal Kengne, Mosa Moshabela, Nasheeta Peer, Tsaone Tamuhla, Nicki Tiffin, Nateiya M. Yongolo, Gift T. Banda, Edna Bosire, Jonathan Chiwanda, Rashida A. Ferrand, Emily Mendenhall, Jamie Rylance, Sangwani Salimu, Felix Limbani.

**Supervision:** Justin Dixon, Ben Morton.

**Writing – original draft:** Justin Dixon, Ben Morton, Misheck J. Nkhata, Alan Silman, Ibrahim G. Simiyu, Stephen A. Spencer, Myrna Van Pinxteren.

**Writing – review & editing:** Christopher Bunn, Claire Calderwood, Clare I. R. Chandler, Amelia C. Crampin, John R. Hurst, Modou Jobe, Andre Pascal Kengne, Naomi S. Levitt, Mosa Moshabela, Mayowa Owolabi, Nasheeta Peer, Nozgechi Phiri, Sally J. Singh, Tsaone Tamuhla, Mandikudza Tembo, Nicki Tiffin, Eve Worrall, Nateiya M. Yongolo, Fanuel Bickton, Rashida A. Ferrand, Felix Limbani.

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
