## [Decision Letter · Decision Letter 0]

13 Mar 2024

PGPH-D-23-02043

Interdisciplinary perspectives on multimorbidity in Africa: developing an expanded conceptual model

Dear Dr. Dixon,

Thank you for submitting your manuscript to PLOS Global Public Health. After careful consideration, we feel that it has merit but does not fully meet PLOS Global Public Health’s publication criteria as it currently stands. Therefore, we invite you to submit a revised version of the manuscript that addresses the points raised during the review process.

We look forward to receiving your revised manuscript.

Kind regards,

Andre F. S. Amaral, Ph.D.

Academic Editor

Journal Requirements:

2. Please provide separate figure files in .tif or .eps format only and remove any figures embedded in your manuscript file. Please also ensure all files are under our size limit of 10MB.

4. Some material included in your submission may be copyrighted. According to PLOS’s copyright policy, authors who use figures or other material (e.g., graphics, clipart, maps) from another author or copyright holder must demonstrate or obtain permission to publish this material under the Creative Commons Attribution 4.0 International (CC BY 4.0) License used by PLOS journals. Please closely review the details of PLOS’s copyright requirements here: PLOS Licenses and Copyright. If you need to request permissions from a copyright holder, you may use PLOS's Copyright Content Permission form.

Potential Copyright Issues:

Fig 1: please (a) provide a direct link to the base layer of the map (i.e., the country or region border shape) and ensure this is also included in the figure legend; and (b) provide a link to the terms of use / license information for the base layer image or shapefile. We cannot publish proprietary or copyrighted maps (e.g. Google Maps, Mapquest) and the terms of use for your map base layer must be compatible with our CC-BY 4.0 license. 

Reviewers' comments:

Reviewer's Responses to Questions

**Comments to the Author**

1. Does this manuscript meet PLOS Global Public Health’s publication criteria? Is the manuscript technically sound, and do the data support the conclusions? The manuscript must describe methodologically and ethically rigorous research with conclusions that are appropriately drawn based on the data presented.

Reviewer #1: Yes

2. Has the statistical analysis been performed appropriately and rigorously?

Reviewer #1: N/A

3. Have the authors made all data underlying the findings in their manuscript fully available (please refer to the Data Availability Statement at the start of the manuscript PDF file)?

Reviewer #1: Yes

4. Is the manuscript presented in an intelligible fashion and written in standard English?

Reviewer #1: Yes

5. Review Comments to the Author

Reviewer #1: Overall summary and impression

In this manuscript the authors present their analysis of data collected at a 3 day workshop focused on multimorbidity in Africa. Data were qualitative in nature and authors were guided by principles of participatory research and grounded theory methodology. The authors’ analysis resulted in the production of 3 thematic areas which broadly highlighted deficits associated with the current definition of multimorbidity (MM) and opportunities for a revised definition. Participants at the workshops suggested a revised definition should be focused on the experiences of those affected, could be used to transform siloed models of health care, and could respond to the needs of various interdisciplinary stakeholders. Following the workshop, the authors produced a revised model of multimorbidity which incorporated both workshop data and the existing literature. The model is also tailored to the African-context.

The manuscript is worthy of publication in PLOS ONE because the authors write about important issues for the future of global health. The methods as described are sufficiently rigorous, and the data from the workshop supports the authors’ conclusions. However, the manuscript could be strengthened with revisions. As articulated below, I suggest the authors focus on revising the text to clarify and condense the text as well as describe the research methods in greater detail.

Major Feedback

1) Methods section needs more information on data collection and analysis. (See minor feedback for specific points)

2) Results section:

a. Needs to include more data on what is unique about the African context (e.g., more single disease-donor funded health care, higher background rates of poverty and higher potential for catastrophic costs compared to high income countries, higher background levels of infectious diseases) in all three thematic areas to answer the research questions.

b. Some sentences seem to be in the wrong paragraphs for their thematic focus and should be rearranged. (e.g., sentences at line 338- 351 can go in theme two; sentences at line 377- 380 should move to theme one)

c. Themes could be named more clearly communicate the focus-- in plain language if possible. For example, theme one may be renamed “inability for current definition to address deficits in care for people living with MM (PLWMM) in Africa”. Theme two may be renamed “potential for revised definitions to centre the experience of living with MM in Africa”. Theme three may be renamed “need for revised definitions to respond to varied conceptual priorities amongst stakeholders”.

d. More sentences in the results section should start with words like “participants stated, clinicians reported, community representatives emphasized, researchers elaborated”. The way it is currently written (with fewer sentences starting this way) makes many sentences in the results section sound like definitive facts rather than the thoughts of the participants. This is especially problematic in theme two as participants are speaking on behalf of PLWMM.

e. The results section would be much more readable, and the credibility of the analysis would be enhanced by moving quotes from boxes into the text to support the relevant statements.

3) The text could be condensed and summarized throughout. The discussion and analytic framework sections are particularly lengthy and could be shortened.

a. The authors do not need to review each thematic area separately in the discussion section. I suggest the authors pull out the main overarching points, which I have interpreted to be:

i) The data in this study supports the conclusions of other authors who have stated that the concept of MM must move away from the biomedical tendency to define people by their disease.

ii) Participants in this study stated that there has been a great deal of person-centred theorizing about the concept of MM, but African health systems have not responded to this literature by providing more holistic care. For example, our results are aligned with other African qualitative researchers who have pointed out that health systems must address patients’ socioeconomic needs to prevent them from tipping into catastrophic consequences.

iii) As seen in our proposed definition, we believe MM and should incorporate concepts from systems, complexity, Ubuntu, and syndemic theories to demonstrate the need for holistic, integrated models of care for PLWMM in Africa. To that end, we have incorporated these theories with workshop data to develop a model depicting intertwined biological, socio-cultural, political, economic, and environmental factors.

Minor Feedback

Introduction

• Lines 152- 157- to better frame research questions there should be a more explicit statement here about elements which are specific to the African context. Elements could include less health funding per capita; more fragmented care due to high proportion of NGO funding; higher background level of communicable disease; PLWMM in Africa are more likely to experience poverty, have little access to a social safety net, and are more likely to experience catastrophic consequences of MM.

• Line 182-188- please include one to two additional sentences on the current controversies with existing MM definitions and what alternative definitions have been suggested.

Methods

• Line 212- further water down claims to following grounded theory methodology as authors did not use intersperse analysis and theoretical sampling but used thematic analysis. I suggest the methodological claim could be re-stated as “This was a qualitative study influenced by principles of participatory research and constructivist grounded theory methodology.” I feel inclusion of the word ‘participatory’ is important because participants contributed to analysis and dissemination.

• Line 225- Text should match diagram, particularly related to what disciplines of clinicians were present (i.e., nursing and physicians). Also missing here is that civil society and community representatives were present. Additional details on what kind of community and civil society representation was present would be helpful.

• Line 259- If it is true that the data in this article limited to the first thematic domain only (i.e., framings and concepts of MM) ,then this should be clearly stated.

• There needs to be a distinct data collection section. The sentences at lines 281- 288 should be moved to the data collection section. Also, information is needed on how the data collectors were instructed to take notes, how detailed the notes were (e.g., were comments summarized by speaker or summarized for the whole group?, how were notes anonymized?, how were notes collated and disseminated amongst participants?)

• Line 292- 299- Could these lines be rephrased to be more explicit and chronological? (For example, first the coding team reviewed all/some of the notes, then there was a meeting amongst the coding team when the initial framework was developed, then XX and YY applied the coding framework to all/some of the notes, had another meeting…)

o Line 292 - Were initial themes identified or initial codes identified?

o Line 295 – Is the final code tree the three themes? I don’t see a coding tree in the results section.

o Line 295-299- I assume this is the process by which the model in the discussion section was developed, but this process could be more clearly described. (For example, first XX and YY consulted the relevant literature, then developed a tentative visual mode

---

## [Editor Report · Decision Letter 1]

13 Jun 2024

Interdisciplinary perspectives on multimorbidity in Africa: developing an expanded conceptual model

PGPH-D-23-02043R1

Dear Dr Dixon,

We are pleased to inform you that your manuscript 'Interdisciplinary perspectives on multimorbidity in Africa: developing an expanded conceptual model' has been provisionally accepted for publication in PLOS Global Public Health.

Best regards,

Andre F. S. Amaral, Ph.D.

Academic Editor